# On the Role of Fixed Points of Dynamical Systems in Training Physics-Informed Neural Networks

**Franz M. Rohrhofer** *frohrhofer@acm.org*
*Know-Center GmbH – Research Center for Data-Driven Business & Big Data Analytics*
*Sandgasse 36/4, 8010 Graz, Austria*

**Stefan Posch** *stefan.posch@lec.tugraz.at*
*LEC GmbH – Large Engines Competence Center*
*Inffeldgasse 19, 8010 Graz, Austria*

**Clemens Gößnitzer** *clemens.goessnitzer@lec.tugraz.at*
*LEC GmbH – Large Engines Competence Center*
*Inffeldgasse 19, 8010 Graz, Austria*

**Bernhard C. Geiger** *geiger@ieee.org*
*Know-Center GmbH – Research Center for Data-Driven Business & Big Data Analytics*
*Sandgasse 36/4, 8010 Graz, Austria*

**Reviewed on OpenReview:** *https://openreview.net/forum?id=56cTmVrg5w*

## Abstract

This paper empirically studies commonly observed training difficulties of Physics-Informed Neural Networks (PINNs) on dynamical systems. Our results indicate that fixed points which are inherent to these systems play a key role in the optimization of the in PINNs embedded physics loss function. We observe that the loss landscape exhibits local optima that are shaped by the presence of fixed points. We find that these local optima contribute to the complexity of the physics loss optimization which can explain common training difficulties and resulting nonphysical predictions. Under certain settings, e.g., initial conditions close to fixed points or long simulations times, we show that those optima can even become better than that of the desired solution.

## 1 Introduction

Dynamical systems are governed by differential equations and are ubiquitous in many scientific disciplines including economics, biology, physics and engineering. The upsurge in scientific machine learning has led to the development of sophisticated approaches, such as Gauss-Markov processes (Schober et al., 2014) or numerical Gaussian processes (Raissi et al., 2018), that are applicable to those systems and superior to classical methods. In the field of deep learning, state-of-the-art methods have advanced by incorporating (at least) some part of the underlying physics, e.g., learned through data (Brunton et al., 2016) or embedded by design (Sanchez-Gonzalez et al., 2020). Among those methods are *physics-informed neural networks* (PINNs) which are the prime paradigm of physics-informed machine learning (Raissi et al., 2019; Karniadakis et al., 2021). Their seamless integration of data and physical constraints has pushed PINNs into a vast number of applications on dynamical systems, including system identification (Raissi, 2018), hidden state inference (Raissi et al., 2020) and surrogate modeling (Sun et al., 2020).

Since PINNs are capable of solving differential equations in a fully mesh-free and time-continuous manner, one promising field of application is the numerical simulation of dynamical systems. In those applications, labeled training data is scarce and typically only used to specify the corresponding initial and boundary conditions (IC/BC). For a complete and unique definition of the forward problem, the IC/BC are either

included in the loss function or explicitly enforced using specific network architectures. Both variants, however, rely on the optimization of the embedded physics loss function, i.e., on minimizing residuals on the governing differential equations, evaluated at collocation points which are randomly sampled inside the computational domain. Optimization success and accuracy thus particularly depend on the complexity of the studied dynamical system and the corresponding physics loss function.

## 1.1 Training Difficulties of Physics-Informed Neural Networks

In general, issues in the optimization of PINNs are manifold and often cause incorrectly predicted system dynamics. A complete description of all reported training difficulties in PINNs is exhaustive; we thus focus our discussion on issues and proposed remedies which appear in the context of dynamical systems and relate to what is discussed in our work.

### 1.1.1 Conflicting Objectives.

Several ongoing discussions address conflicts in the optimization of multiple objectives as one root cause of convergence issues in PINNs (Wang et al., 2021a). It has been shown that different weighting of the objectives, bound to physical, IC/BC and data constraints, is effective for PINNs and can accomplish a successful training. The weighting is conducted either by hand-tuned loss weights or with adaptive weighting schemes that adjust the weights during network training, such as in Maddu et al. (2021) or Jin et al. (2021). With a focus on coping with imbalanced gradients, those methods are generally used to improve the PINN's performance and to select an optimum point on the Pareto front (Rohrhofer et al., 2021). Furthermore, specially-designed network architectures enable hard encoding of IC/BC and physical constraints (Lu et al., 2021; Raissi et al., 2019). These approaches circumvent conflicts by reducing the overall number of competing objectives.

### 1.1.2 Propagation Failure.

Most relevant for our discussion are recent works that focus on a purported failure mode of PINNs in which the learned system dynamics does not represent the solution that is specified by the IC/BC. A reason for this, it has been argued, is that propagation of the solution from the enforced conditions to interior points is disrupted for a certain region in the computational domain, which often yields the trivial (zero) solution (Daw et al., 2022). To mitigate this issue, several remedies have been proposed. One focus lies in improving network initializations to reduce the bias towards flat output functions, e.g., by learning in sinusoidal space (Wong et al., 2021). Another type of methods propose reweighting of collocation points (Wang et al., 2022) or resampling them during network training (Leiteritz & Pflüger, 2021) . In those methods, importance or density of collocation points propagates from the enforced conditions to interior points during network training which, in a causality-respecting manner, promises to mitigate the propagation failure. Since it is generally argued that with an increasing domain size the physics loss optimization becomes more complex (Krishnapriyan et al., 2021), other methods focus on the extent of the computational domain. Often referred to as sequence-to-sequence learning or domain decomposition, those methods comprise approaches that divide the original spatio-temporal domain into smaller subdomains which are easier to solve (Jagtap & Karniadakis, 2021). Furthermore, approximation issues of PINNs in the presence of high-frequency or multi-scale features have been explained by the spectral bias of PINNs with proposed remedies found in Wang et al. (2021b).

## 1.2 Our Contribution

As shown in the last section, the literature is abound with techniques that try to mitigate commonly observed training difficulties of PINNs – but explanations why training on dynamical systems often fails seems incomplete to us: We suspect that not only the trivial zero solution, but also fundamental properties, e.g., fixed points of dynamical systems play a key role in training (failures) of PINNs. Based on this, our contribution in this paper will be as follows.

- We show on two simple dynamical systems that stable *and* unstable fixed points contribute to the optimization complexity of the physics loss function and influence the rate of training success. (Section 3.1.1)

- We empirically demonstrate that under certain settings, e.g., IC close to fixed points and long simulation times, nonphysical predictions become economical with better minima than that of the desired solution. (Section 3.1.2)

- We further demonstrate that PINN training for complex dynamical systems is also affected by fixed points inherent to these systems. (Section 3.2)

- We visually capture that the physics loss landscape is being shaped by the presence of fixed points, which form local optima / saddle points that slow down the gradient-based optimization and might prevent a successful PINN training. (Section 3.3)

- We provide empirical evidence that the complexity of the physics loss landscape reduces for smaller computational domains and connect this to successful techniques used for training PINNs. (Sections 3.3 and 4)

## 2 Background

### 2.1 Dynamical Systems

In this work we consider dynamical systems that can be described by differential equations of the form:

$$u_t = \mathcal{F}[u], \tag{1}$$

where the solution function $u = u(t, x)$ in general depends on time $t \in [0, T]$ and space $x \in \Omega \subseteq \mathbb{R}^n$, $u_t$ denotes the (partial) derivative of $u$ w.r.t. time, and $\mathcal{F}$ is an arbitrary, potentially nonlinear differential operator dictating the system dynamics. In the numerical simulation of dynamical systems, IC are imposed to define the initial state of the system through

$$u(t_0 = 0, x) = u_0(x), \quad \forall x \in \Omega. \tag{2}$$

For the dynamical systems considered in this work, the spatial domain $\Omega$ is compact, i.e. closed and bounded, which further requires the specification of BC on the boundary $\partial\Omega$ in order to guarantee the uniqueness of the solution:

$$\mathcal{B}[u] = 0, \quad \forall x \in \partial\Omega, \tag{3}$$

where $\mathcal{B}$ is a boundary operator, specifying periodic, Dirichlet and/or Neumann BC.

### 2.2 Physics-Informed Neural Networks

Fully-connected neural networks (FC-NNs) are most common among PINNs due to their good trade-off between simplicity and expressive power. Thus, we use FC-NNs to approximate the unknown solution function of (1) with $u(t, x) \approx u(t, x; \theta) =: u_\theta(t, x)$, where $\theta \in \mathbb{R}^{n_\theta}$ are the weights and bias terms of the network. Common activation functions are the hyperbolic tangent (tanh) or Sigmoid linear unit (SiLU, swish), which render the approximated solution function and derivatives smooth[1].

PINNs use automatic differentiation (AD) (Baydin et al., 2018) to obtain (partial) derivatives of the network's output with respect to its inputs. In order to retrieve the derivatives of the network solution, AD requires to pass discrete evaluation points, called collocation points, through the network in a feed-forward operation. The collocation points define the data set for penalizing residuals of the differential equation. The physics loss residual for a dynamical system (1) is given by:

$$f(t, x) := u_{\theta, t}(t, x) - \mathcal{F}[u_\theta(t, x)], \tag{4}$$

---

[1]Thus mesh-free and time-continuous in the context of differential equations.

where we use $u_{\theta,t}$ to denote the derivative of the network function $u_\theta$ w.r.t time $t$. Following the standard PINN formulation, the sum of squared residuals at all collocation points yields the physics loss function:

$$L_f(\theta) = \frac{1}{N_f} \sum_{i=1}^{N_f} \left| f(t^i, x^i) \right|^2, \tag{5}$$

with the collocation points $\{t^i, x^i\}_{i=1}^{N_f}$ sampled from the entire computational domain $(t, x) \in [0, T] \times \Omega$. These points do not need any label and can be either fixed during PINN training or re-sampled before each training epoch. In the initial formulation of PINNs, additional data constraints are used to enforce the IC/BC by

$$L_u(\theta) = \frac{1}{N_u} \sum_{i=1}^{N_u} \left| u_\theta(t^i, x^i) - u(t^i, x^i) \right|^2, \tag{6}$$

where $u$ is given by the r.h.s. of (2) and (3). Both losses (5) and (6) are combined by scalarization of a multi-objective optimization through

$$L(\theta) = \lambda L_u(\theta) + L_f(\theta), \tag{7}$$

where $\lambda$ represents a weighting factor, which here by default is set to $\lambda = 1$, unless explicitly stated otherwise. As an alternative to this (standard) formulation of PINNs, special network architectures have been proposed that ensure IC/BC are satisfied explicitly. We refer to these PINNs as being *hard constrained*.

## 2.3 Stability, Fixed Points & Steady-State

Fixed points $u^*$ of a dynamical system are given by the roots of the nonlinear function $\mathcal{F}$ in equation (1):

$$\mathcal{F}[u^*] = 0. \tag{8}$$

In general, they can be either stable, asymptotically stable or unstable. For a stable fixed point, any trajectory close to it will stay close, whereas for an asymptotically stable fixed point close trajectories will further converge to it as $t \to \infty$. In contrast, an unstable fixed point is repulsive and even the smallest deviation will cause any close trajectory move away from it as $t \to \infty$. While it is hard to determine all fixed points and their asymptotic properties in dynamical systems, the trivial zero solution $u^* = 0$ is a fixed point for many systems, such as for harmonic oscillators, pendulums or fluid flow.

In the context of ordinary differential equations (ODEs) fixed points of dynamical systems are constant solutions which are often studied from the perspective of stability, to characterize the asymptotic properties of solutions and trajectories close to it. For dynamical systems governed by partial differential equations (PDEs), fixed points appear in obtaining certain solutions to a given system. Some applications aim at obtaining steady-state solutions, i.e. solutions that do not change over time as the state variables are changed and, thus, for which equation (8) holds true. When seeking those solutions, it is the asymptotic property of an underlying stable fixed point which determines what happens to the system in the long term after it has been initiated. Other applications, however, consider the repulsive property of an unstable fixed point, to bring the system out from steady-state and to obtain a transient solution. An particular example for the latter is simulating vortex shedding which appears in fluid dynamics and will be also studied in this work.

## 3 On the Attractivity of Fixed Points in Physics-Informed Neural Networks

According to (8), if the parameters $\theta$ of the PINN are such that $u_\theta = u^*$ corresponds to a fixed point, then $\mathcal{F}[u_\theta] = 0$ and the physics loss in (4) vanishes. Further, for PINN architectures with finitely many layers and neurons per layer, and for continuous activation functions, the network solution $u_\theta$ changes continuously with $\theta$. Thus, if $\theta'$ is a sufficiently small perturbation of $\theta$, $u'_\theta$ is approximately a fixed point solution, i.e., $\mathcal{F}[u'_\theta] \approx 0$. As a consequence, the physics loss for $u'_\theta$ will be positive, but small. Hence, fixed points correspond to global minima of the physics loss with non-trivial basins of attraction. The PINN can thus only learn "correct" behavior from the given IC/BC data. Regardless of whether this IC/BC data is incorporated

using hard constraints or a loss term such as (6), the resulting loss landscape seen by the gradient-based optimizer will be affected by the fixed point solution and its basin of attraction.

In this section we now perform experiments[2] to test the hypothesis that fixed points play a key role in the training of PINNs. In the first part, we perform experiments on two dynamical systems governed by ODEs which are the undamped pendulum and a simple toy example. Due to their simplicity, those systems are studied in multiple settings, including the variation of IC, simulations length, network and optimizer settings. In the second part, we show that our findings are also applicable to complex dynamical systems which are governed by PDEs. We perform experiments on two complex dynamical systems which are the Navier-Stokes equations and the Allen-Cahn equation. Those systems are known to be challenging for PINNs and we relate commonly observed training difficulties to fixed points inherent to these systems.

## 3.1 Fixed Points in Ordinary Differential Equations

In the following, we consider two simple ODEs: the undamped pendulum dynamics and a simple toy example. These examples are chosen because they exhibit stable (pendulum), asymptotically stable (toy example), and unstable (both) fixed points. We further try to solve these systems using either vanilla (i.e., multi-objective, for the pendulum) and hard constrained (for the toy example) PINNs. For both examples we now stick to the convention of ODEs and denote the unknown solution function by $y(t)$.

**Undamped Pendulum Equation.** The undamped pendulum dynamics are given by a second-order ODE

$$\ddot{y} = \mathcal{F}[y] := -\frac{g}{l}\sin\left(y\right),\tag{9}$$

with $y$ denoting the angle from the vertical to the pendulum (in radians), and $l$ and $g$ representing the length of the rod and magnitude of the gravitational field, respectively. Here we set $l = 1$ and $g = 9.81$ and consider for our discussion the angle $y$ in units of degrees.

The second-order ODE (9) can be converted into a coupled systems of two first-order ODEs which are in the form of equation (1). Since PINNs are able to also handle higher-order differential equations, we use a single neural network $y_\theta(t)$ to approximate the solution function and solve the second-order ODE.

This system exhibits two fixed points, a stable fixed point $y^* = 0°$ at the pendulum's natural rest position, and an unstable fixed point $y^* = 180°$ at the upright position. For comparison in our experiments, we create reference solutions using a Runge-Kutta fourth-order method.

**Toy Example Equation.** The toy example is defined as a one-dimensional system with a single ODE given by

$$\dot{y} = \mathcal{F}[y] := y\left(1 - y^2\right).\tag{10}$$

This system exhibits three fixed points, two of which located at $y^* = \pm 1$ are asymptotically stable, and one at $y^* = 0$ which is unstable (see Figure 1). For this system, the analytical solution exists and is provided in Appendix C.1. The simplicity of this toy example further allows for hard constraining the IC by setting

$$\hat{y}(t) = y_0 + t \cdot y_\theta(t),\tag{11}$$

which per definition fulfills equation (2).

### 3.1.1 Rate of Training Success in the Presence of Fixed Points

We now study the success rate of training PINNs on the above mentioned systems. We declare training successful if the $L_2$ relative error $\|y_\theta(t) - y(t)\|_2 / \|y(t)\|_2$ is below 15%, when comparing the PINN's prediction with the reference solution after a sufficiently long training time. This threshold allows a clear separation of the training outcomes[3].

For the toy example nonphysical predictions are exclusively influenced by the unstable fixed point at $y^* = 0$ (see Figure 1(c)). For the undamped pendulum, however, we further classify whether an unsuccessful training,

---

[2]Code can be found on GitHub at `https://github.com/frohrhofer/PINNs_fixed_points`

[3]Results for further thresholds are provided in Appendix B.3 and suggest similar qualitative conclusions.

Table 1: **Undamped Pendulum.** Rate of training success across different system settings ($T$ and $y_0$) and network architectures (size and activation function). Triplets in the main table represent in percentage (%) and in the respective order, cases of successful training, attracted by stable fixed point, and unstable fixed point (see Appendix B.1 for details on the classification). Bold triplets represent a low ($< 5\%$) success rate.

| $T$ | | 2.5 | | | 5 | | | 7.5 | | |
|---|---|---|---|---|---|---|---|---|---|---|
| $y_0$ | | $25°$ | $100°$ | $175°$ | $25°$ | $100°$ | $175°$ | $25°$ | $100°$ | $175°$ |
| | tanh | 98/2/0 | 100/0/0 | 100/0/0 | **0/100/0** | 90/10/0 | **0/100/0** | **0/100/0** | **0/100/0** | **0/61/39** |
| 4x50 | swish | 100/0/0 | 100/0/0 | 98/0/2 | 56/44/0 | 80/20/0 | **0/100/0** | **0/100/0** | **1/99/0** | **0/91/9** |
| | sin | 100/0/0 | 100/0/0 | 100/0/0 | 65/35/0 | 93/7/0 | **0/93/7** | **2/98/0** | 24/75/1 | **0/94/6** |
| | tanh | 100/0/0 | 100/0/0 | 99/0/1 | 100/0/0 | 97/0/3 | 92/0/8 | 49/31/20 | 84/0/16 | **1/29/70** |
| 8x100 | swish | 100/0/0 | 100/0/0 | 100/0/0 | 100/0/0 | 100/0/0 | 57/42/1 | 100/0/0 | 100/0/0 | **1/92/7** |
| | sin | 100/0/0 | 100/0/0 | 98/0/2 | 55/0/45 | 60/0/40 | 39/15/46 | 32/0/68 | 29/0/71 | **0/26/74** |

subsequently the nonphysical prediction, violates the physics by either being attracted by the stable ($y^* = 0°$) or the unstable fixed point ($y^* = 180°$). This classification is based on $y$ and $\dot{y}$, i.e., in phase space, evaluated at $t = T$ (see Appendix B.1 for details).

For both systems, we train 100 randomly initialized PINNs with a different IC $y_0$ and simulation time $T$. The IC for the undamped pendulum are enforced using the multi-objective loss (7) (vanilla PINN) with a zero initial angular velocity, i.e., $\dot{y}_0 = 0$. Training is performed for 50k epochs using the Adam optimizer with default settings for the moment estimates. We repeat training for different setups, including network architectures and optimizer settings (see Appendix B.2 for details). Software and hardware specifications in use are found in Appendix A.

Table 1 shows the rate of training success across different network architectures for the experiments on the undamped pendulum and an initial learning rate of $\alpha = 0.001$. The outcome of experiments using different optimization settings and for the toy example can be found in Appendix B.2 and C.2. In general, we observe severe training difficulties across all experiments with only a minor number of cases leading to a success rate of 100%.

**Influence of Initial Condition $y_0$.** From the results it is apparent that PINNs are sensitive to the choice of initial conditions. We observe that, in general, IC close to fixed points lead to a lower rate of training success compared to those that start far. For the undamped pendulum this is evident by comparing in Table 1 the rates for $y_0 = 100°$ with that of $y_0 = 25°$ and $y_0 = 175°$. We made similar observations for the toy example that also shows a lower rate of success for IC close to the unstable fixed point (see Appendix C.2).

**Influence of Simulation Time $T$.** Across different settings and for both systems, we observe that training becomes more difficult as the simulation time is increased. Likewise it can be seen that reducing the simulation time accounts for a higher success rate. The influence of the simulation time in terms of the physics loss complexity will be further analyzed in Section 3.3.

**Influence of Network and Optimization Settings.** We observe a slight improvement in terms of a higher success rate as the network size is increased. Still, none of the tested network architectures could resolve the training issues at long simulation times and IC close to the (unstable) fixed point. Thus, we conclude that the observed training difficulties are bound to the optimization complexity, rather than insufficient expressive power of the network. As reported in Appendix B.2, we also perform an ablation study using different optimization settings in terms of the learning rate, number of collocation points, loss weighting and network initialization. In comparison to the baseline model (4x50, tanh, from Table 1) no optimization setting yields notable changes to the rate of training success.

### 3.1.2 Fixed Points Becoming Economical Solutions

Next, we demonstrate that, when the IC is very close to a fixed point, training may result in PINN predictions that approach these fixed points, even if the resulting behavior is nonphysical. Furthermore, we show that those nonphysical predictions can even become better minima than that of the desired optimum. The

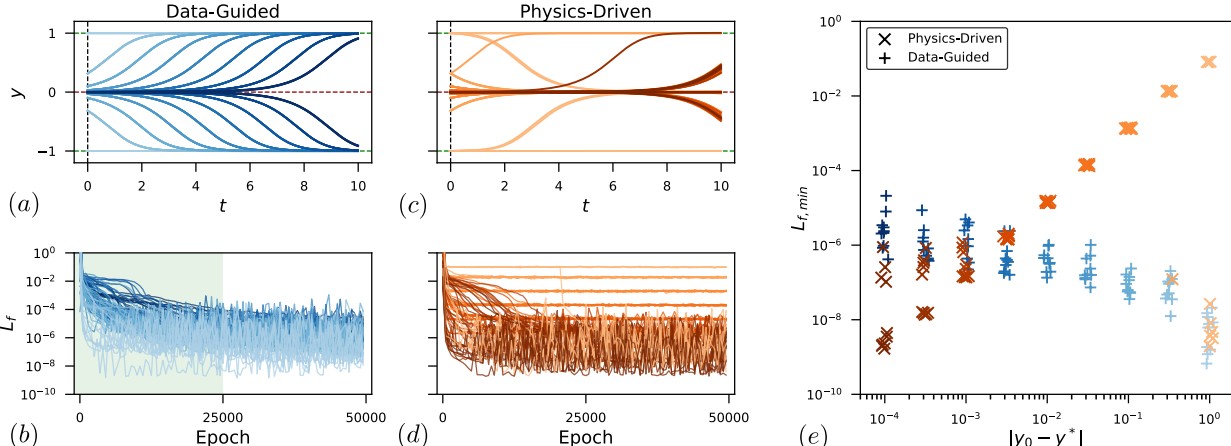

Figure 1: **Toy Example.** Predicted trajectories and learning curves for the $(a)$,$(b)$ data-guided PINN and $(c)$,$(d)$ physics-driven PINN. $(e)$ The minimal physics loss across all epochs vs. absolute distance between IC and the unstable fixed point indicate that nonphysical predictions become better minima than the true solution. Five PINN instances were trained for each initial value (sequential color code) and approach (crosses and pluses). Markers were randomly shifted horizontally to reduce overlap

latter further renders convergence to the true solution unfeasible. To rule out effects from multi-objective optimization, we focus in this subsection on the toy example implemented by a PINN with hard constraints. Similar observations on the undamped pendulum can be found in Appendix B.4.

For this experiment, we use a 4x50 network architecture with tanh activation and choose a simulation time of $T = 10$. To show that our chosen PINN architecture has sufficient expressive power to learn the true solution, we implemented the following, *data-guided* PINN as control: First, 10 labeled data points are sampled from the analytical solution, equidistantly from the computational domain. In the first 25k epochs of training, these 10 points support training via (7), while in the remaining 25k epochs, the PINN is trained using the physics loss only. The first training phase thus guides the gradient-based optimization into the basin of attraction of the analytical solution, while the second training phase ensures that the physics loss is minimized. Indeed, as shown in Figure 1$(b)$, the data-guided strategy successfully learns the analytical solution. We compare this approach to training with the physics loss only, i.e., without the use of labeled data, which we refer to as the *physics-driven* PINN. For each approach and IC we train five uniquely initialized PINN instances with the same optimization settings found in 3.1.1.

**Influence of Initial Condition $y_0$.** For the toy example, Figure 1$(c)$-$(e)$ show that the PINN predictions impacted by the unstable fixed point ($y^* = 0$) achieve lower physics losses as the IC gets closer to it. Indeed, for small values of $y_0$, the physics loss can become even smaller than the physics loss achieved by the data-guided PINN (see Figure 1$(e)$), i.e., the prediction impacted by the unstable fixed point seems to become a better minimum for PINN optimization than the true solution. This renders finding the true solution unfeasible, even for optimization methods that are not based on gradients (e.g., PSO-PINNs as in Davi & Braga-Neto (2022) or else).

### 3.2 Fixed Points in Partial Differential Equations

In the following, we broaden our discussion on two complex dynamical systems which are governed by PDEs: the Navier-Stokes equations and Allen-Cahn equation. We show that common training difficulties and frequently-observed nonphysical predictions on these systems can be explained by the presence of fixed points inherent to the experimental setups.

**Navier-Stokes Equations.** The Navier-Stokes equations govern fluid flow and build a coupled system of nonlinear PDEs. For the two-dimensional case, the unknown solution functions are $u(t, \vec{x})$, $v(t, \vec{x})$ and $p(t, \vec{x})$, representing the fluid velocity in $x$- and $y$-direction and pressure, respectively. The system equations for transient fluid flow, representing conservation of momentum in $x$- and $y$-direction, are given by

$$u_t = \mathcal{F}_x[u, v, p] := -(uu_x + vu_y) - p_x + \mathrm{Re}^{-1}(u_{xx} + u_{yy}), \tag{12a}$$

$$v_t = \mathcal{F}_y[u, v, p] := -(uv_x + vv_y) - p_y + \mathrm{Re}^{-1}(v_{xx} + v_{yy}). \tag{12b}$$

In this work we consider incompressibility of the fluid which is given by the continuity equation

$$u_x + v_y = 0. \tag{13}$$

In order to hard constrain this additional PDE, we introduce a stream function $\psi(t, \vec{x})$ with $u = \psi_y$ and $v = -\psi_x$, similar to what has been done in Raissi et al. (2019). Consequently, a single neural network can be used to approximate $\psi_\theta(t, \vec{x})$ and $p_\theta(t, \vec{x})$, which per definition fulfills the continuity equation (13).

**Allen-Cahn Equation.** The Allen-Cahn equation describes a reaction-diffusion systems which is used to simulate phase separation in multi-component alloy systems. The highly non-linear PDE is given by

$$u_t = \mathcal{F}[u] := \gamma_1 u_{xx} + \gamma_2 (u - u^3). \tag{14}$$

The following proposition (proof is given in Appendix E) characterizes a non-trivial fixed point inherent to this system:

**Proposition 1.** *Consider a PINN instance, where the collocation points are drawn from a continuous distribution supported on the computational domain $[-1, 1] \times [0, T]$, and suppose that the PINN is used to approximate the solution to the Allen-Cahn equation (14). Then, for the function*

$$u(x, t) = \begin{cases} 0, & x \in [-0.5, 0.5] \\ -1, & x \in [-1, -0.5) \cup (0.5, 1] \end{cases} \tag{15}$$

*the physics loss $L_f(\theta) = 0$ with probability one.*

### 3.2.1 Fixed Point Leading to Steady-State

For this experiment we consider the Navier-Stokes system (12) and aim at simulating vortex shedding, which is a well-studied phenomenon and benchmark simulation in fluid dynamics. The simulation aims at capturing the characteristic properties of oscillating fluid flow, which appears in the wake of a (round) body that is passed by the fluid at a certain velocity. Of particular interest, in terms of stability, is that this transient and periodic flow is obtained by the influence of a fixed point, which becomes unstable above a critical Reynolds number $\mathrm{Re}_c \approx 48$ (Tang & Aubry, 1997). The symmetry breaking, usually achieved by numerical instabilities in classical methods, pushes the symmetrical fluid flow out from steady-state, which conforms to the unstable fixed point, and into the oscillating vortex street.

**Experimental Setup.** The experimental setup for simulating vortex shedding can be found in Appendix D. We choose $\mathrm{Re} = 100$, thus a Reynolds Number high enough to cause the vortex shedding. A database of direct numerical simulation data (DNS) for this Reynolds number can be found in Boudina (2021). This database contains the developed vortex shedding flow field and will be be used as training data for our experiment. We choose a fixed 8x100 neural network architecture with tanh activation and consider following (two) training strategies: One PINN instance is used as control and trained on three consecutive periods of vortex shedding, i.e., $t \in [0, 18]$, with collocation points sampled from the same time domain. We refer to this as *data-guided* PINN, which thus is fully supported by training data and only ask to correctly learn the reference solution. The second *physics-driven* PINN, however, is trained on labeled data coming only from the time domain $t \in [0, 3]$, which represents 50% of the first vortex shedding period. This should impose the initial sequence of vortex shedding, and by sampling collocation points in the full-time domain ($t \in [0, 18]$), the physics-driven PINN is asked to continue the simulation beyond the domain of reference data by minimizing residuals of the Navier-Stokes equations (12). Further details on optimization and data settings can be found in Appendix D.

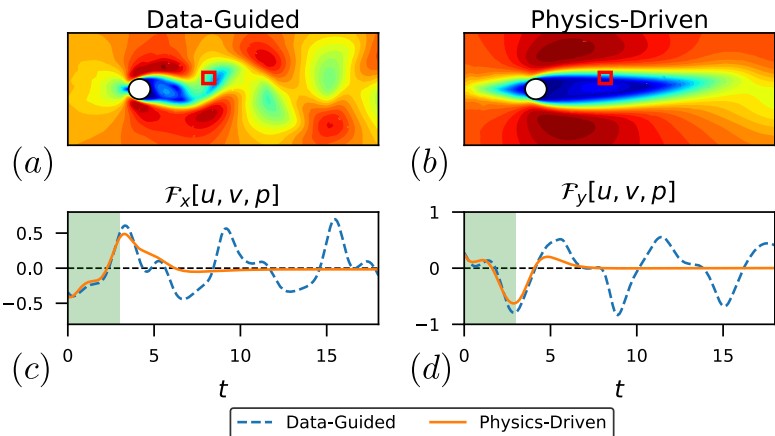

Figure 2: **Navier-Stokes Equations.** Substantial difference in velocity magnitudes $|\vec{u}|$ at $t = 12$ predicted by the $(a)$ data-guided and $(b)$ physics-driven PINN. $(c)$-$(d)$ The time evolution of the nonlinear operators $\mathcal{F}_x$ and $\mathcal{F}_y$ at $(x, y) = (3.0, 0.5)$ (red rectangle) indicates that the physics-driven PINN is influenced by the unstable fixed point which, apparently as $\mathcal{F}_x \to 0$ and $\mathcal{F}_y \to 0$ as $t$ becomes large, attracts a steady-state solution. The green shaded area represents the initial training sequence for the physics-driven PINN.

**Results.** Figure 2 shows for the fully-trained data-guided and physics-driven PINN the predicted velocity magnitudes $|\vec{u}|$ at $t = 12$, and the time evolution of the nonlinear operators $\mathcal{F}_x$ and $\mathcal{F}_y$ at a depicted spatial coordinate $(x, y) = (3.0, 0.5)$. The data-guided PINN indeed resembles the true vortex shedding dynamics (within a certain accuracy not given), which verifies that the network architecture is sufficient large to learn the vortex shedding dynamics over the given time domain. For the physics-driven PINN, however, we observe that outside the domain of reference data the prediction starts to substantially deviate from that of the data-guided. The time evolution of the nonlinear operators $\mathcal{F}_x$ and $\mathcal{F}_y$ gives further insights, and indicates that the physics-driven PINN approaches a steady-state solution with the fixed point properties (8), as it is evident by $\mathcal{F}_x \to 0$ and $\mathcal{F}_y \to 0$ as $t$ becomes large. While this behavior is only shown for a particular spatial coordinate, similar observations are made on the entire spatial domain (not shown). These observations, together with the fact that the approached solution (Figure 2$(b)$) is not the trivial zero solution, suggests that the training is influenced by the underlying unstable fixed point, which originally accounts for the vortex shedding dynamics. We note that evaluating the physics loss for both PINN instances shows that the minimal loss achieved by the physics-driven PINN ($L_{f,\min} = 8.28 \cdot 10^{-5}$) is lower than that of the data-guided PINN ($L_{f,\min} = 2.07 \cdot 10^{-4}$). We further note that we have also tested different network and optimization settings, as well as (hard constrained) BCs and different computational domains, but we were not able to achieve, for any of these settings using standard PINNs, a different qualitative behavior as shown here. The challenge of this optimization problem was similarly reported in Chuang & Barba (2022) and can be resolved, e.g., by using truncated Fourier decomposition with PINNs (Raynaud et al., 2022).

### 3.2.2 Fixed Point Slowing Down Convergence

For this experiment we consider the Allen-Cahn equation (14) and define the IC and periodic BC as

$$u(0, x) = x^2 \cos(\pi x), \quad x \in [-1, 1], \quad t \in [0, 1], \tag{16a}$$
$$u(t, 1) = u(t, -1), \tag{16b}$$
$$u_x(t, 1) = u_x(t, -1), \tag{16c}$$

where we set $\gamma_1 = 0.0001$ and $\gamma_2 = 5$ in equation (14). This particular example was also studied in Mattey & Ghosh (2022) and Wight & Zhao (2020) and showed severe training difficulties for PINNs. In both works, the standard PINN approach resulted in ignoring the IC and learning the trivial zero solution. While a weighted PINN with $\lambda = 100$ in Wight & Zhao (2020) showed a slight improvement (by at least learning the

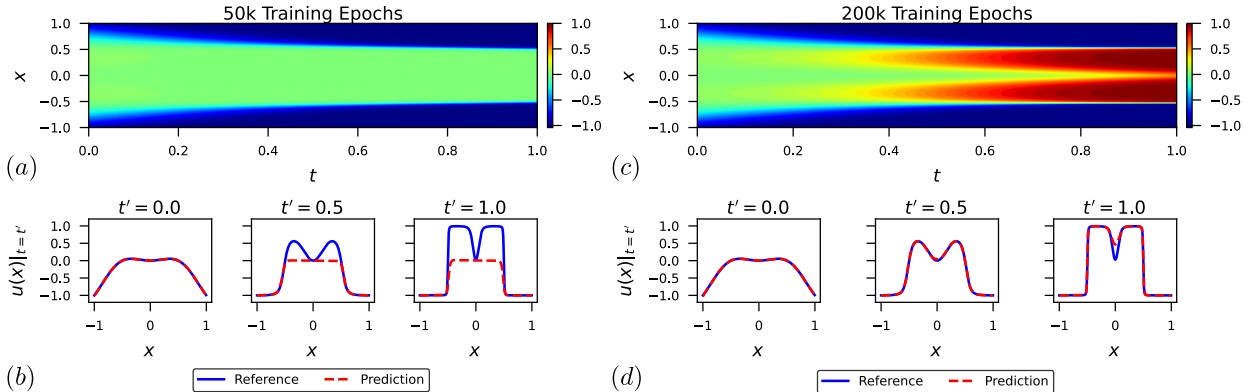

Figure 3: **Allen-Cahn Equation.** Prediction of the PINN trained with ($a$) 50k and ($c$) 200k epochs using gradient-based optimization. ($b$), ($d$) While the prediction of the PINN trained with 200k epochs is in good agreement with the reference, the PINN trained with only 50k epochs is still trapped at a solution which resembles for larger values of $t$ the non-trivial fixed point 15. This suggests that the training is influenced by the fixed point that seems to slow down the gradient-based optimization (will be further analyzed in Section 3.3.2).

IC correctly), both works eventually propose methods that mitigate the observed training difficulties and fall into the category of domain decomposing / adaptive collocation point sampling. The reference solution to this experimental setup can be found in Raissi et al. (2019).

**Experimental Setup.** For this example, we adopt the weighted PINN approach and choose a loss weight $\lambda = 100$ to put greater weight on the IC. We consider a 6x100 network and optimize the composite loss function that enforces the (weighted) IC, BC, and physics loss residuals on equation (14) on the full time domain $t \in [0, 1]$. We train one PINN instance using Adam with 50k epochs and a second instance with a much longer training duration of 200k epochs, both with an initial learning rate of $\alpha = 0.001$. At each training epoch, data for the IC/BC and collocation points are sampled anew with sizes $N_{IC} = 128$, $N_{BC} = 128$ and $N_{col} = 1024$, respectively.

**Results.** Figure 3 shows the prediction of both instances on the entire spatio-temporal domain and on selected time steps. We observe that while both instances correctly learn the IC (and BC), the PINN which is trained on only 50k epochs approaches a solution that resembles the non-trivial fixed point given by (15). We note that these results coincide with that shown in Wight & Zhao (2020), although we used different network and optimization settings. To our surprise, the PINN instance that continues the training for another 150k epochs successfully manages to escape from this suboptimal solution and converges to a solution that is in good agreement with the reference (with minor differences in the late time domain). This observation again suggests that the training is influenced by the non-trivial fixed point (15) that seems to slow down the gradient-based optimization. This particular example, together with the corresponding learning curves, will be further analyzed in the next section.

## 3.3   Optimization Landscape in the Presence of Fixed Points

Finally, we investigate the effect of fixed points on the physics loss landscape. In particular, we evaluate for the toy example and Allen-Cahn system the physics loss function on solutions that were found by the PINN instances and influenced by (close) fixed points. We also assess the effects of reducing the simulation length $T$ by limiting the time domain up to which collocation points are sampled and used to evaluate the physics loss (5).

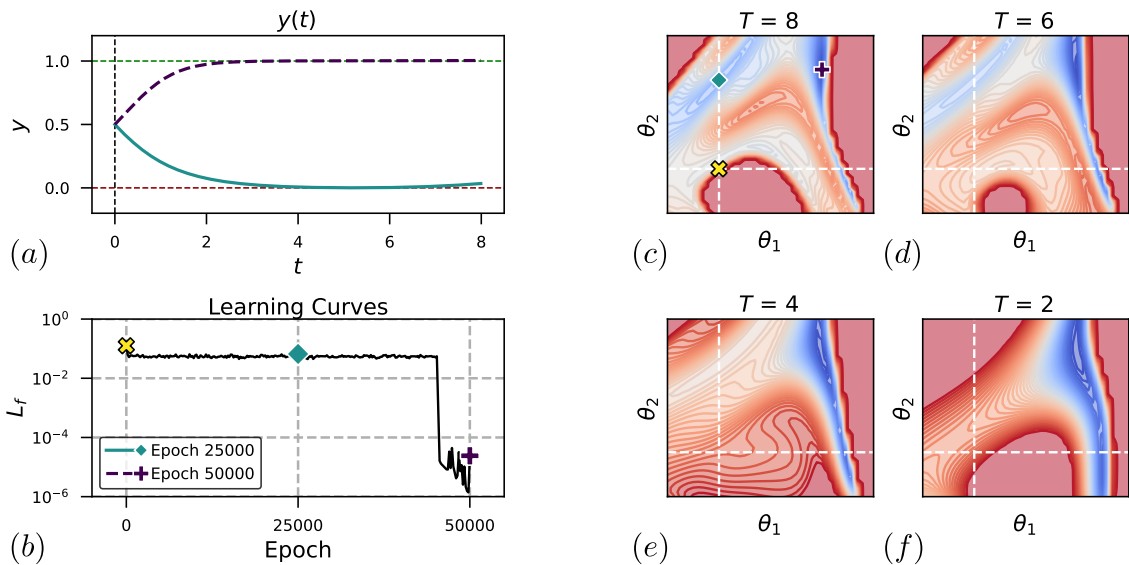

Figure 4: **Toy Example.** $(a)$ PINN prediction and $(b)$ corresponding learning curves for a training example that barely manages to escape from its suboptimal location. $(c)$-$(f)$ Physics loss landscape evaluated on collocation points which are sampled up to $T$. The intermediate nonphysical prediction (diamond) clearly forms an attractive location in the loss landscape, which gradually disappears as $T$ is reduced.

### 3.3.1 Initial Condition Far From Fixed Point

For the toy example in Figure 1$(d)$ we observe that most PINN instances suffer from slow convergence, indicated by the presence of plateaus in the learning curves. Only a few examples show a successful escape from those undesired optima which is evident by a distinct drop in the learning curves. Those cases are primarily reported at IC that are comparably far from the unstable fixed point at $y^* = 0$. To obtain a better understanding of this phenomenon, we plot the physics loss landscape and PINN prediction for a instance ($y_0 = 0.5$ and $T = 8$) that barely manages to escape from its suboptimal location. The two plot directions $\theta_1$ and $\theta_2$ are particularly chosen to point from the initial network state ($\theta^0$) to an intermediate ($\theta^{25k}$) and the final state ($\theta^{50k}$) of the network training. Further visualization details are given in Appendix F.

**Results.** Figure 4 shows the loss landscape for two directions $\theta_1$ and $\theta_2$, where panel $(c)$ represents the loss landscape seen during the PINN optimization. Additionally, in panel $(a)$ and $(b)$ a prediction and training sequence of the PINN is given, which shows that the gradient-based optimization first gets trapped in a local optimum. After approximately 42k epochs, the optimization manages to converge to the correct solution. We clearly detect the global minimum, which corresponds to this solution, in the upper right region (plus) of the loss landscape. We further observe that for long simulation times, i.e. in panel $(c)$-$(e)$, a local minimum or saddle point (diamond) forms, which seems to be attractive to the gradient-based optimization (cf. Figure 1) and apparently conforms to the nonphysical prediction approaching the unstable fixed point. With decreasing $T$, i.e. with a reduced time domain on which the physics loss is evaluated, this local optimum gradually vanishes and the global optimum becomes easier to reach due to the better (in terms of optimization) shape of the loss landscape. This explains why we observe in Table 4 a high rate of training success when using $T = 2.5$.

### 3.3.2 Initial Condition Close To Fixed Point

For our final investigation, we use the PINN instance from the Allen-Cahn system which initially was also trapped at a nonphysical prediction and finally escaped from it as the training continued (cf. Figure 3). In this example, the IC given by (16a) and the resultant dynamics closely pass the fixed point (15) which seemed to affect the PINN training. The PINN was optimized using a composite loss function (7) with

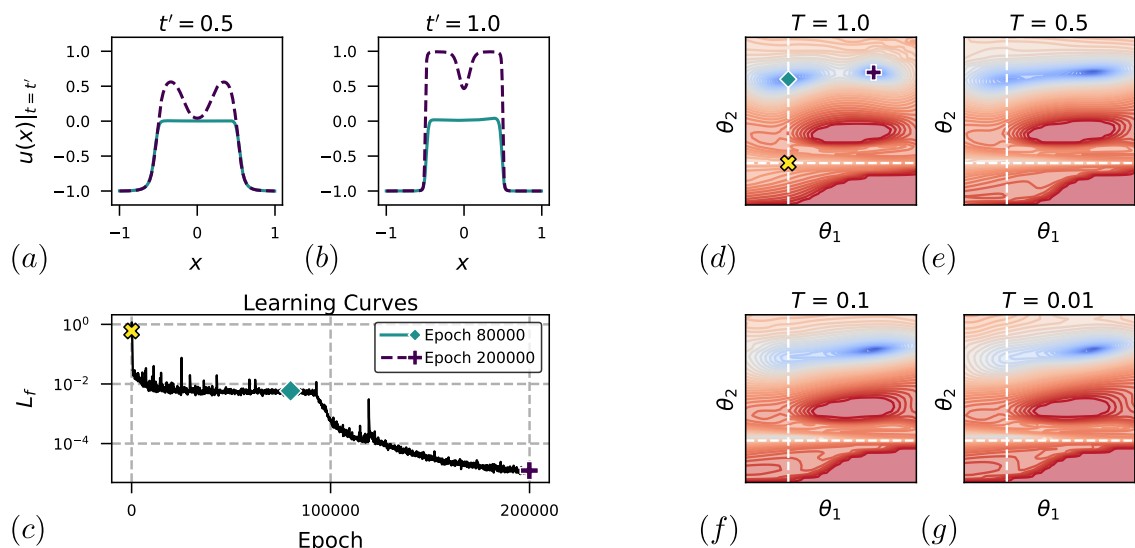

Figure 5: **Allen-Cahn Example.** $(a),(b)$ PINN prediction and $(c)$ corresponding learning curves for a training example that barely manages to escape from its suboptimal location. $(f)$-$(g)$ Physics loss landscape evaluated on collocation points which are sampled up to $T$. The intermediate nonphysical prediction (diamond) clearly forms an attractive location in the loss landscape, which disappears as $T$ is reduced significantly.

$\lambda = 100$. However, since the IC and BC were learned correctly, we only focus on visualizing the physics loss landscape. We plot the loss landscape in two particular directions $\theta_1$ and $\theta_2$ that point from the initial network state ($\theta^0$) to an intermediate ($\theta^{80k}$) and the final state ($\theta^{200k}$) of the network training. Further visualization details are given in Appendix F.

**Results.** Figure 5 shows the loss landscape for two directions $\theta_1$ and $\theta_2$, where panel $(d)$ represents the loss landscape seen during the PINN optimization. Additionally, in panel $(a),(b)$ and $(c)$ predictions and the training sequence of the PINN is given, which shows that the gradient-based optimization first gets trapped in a local optimum (diamond). We note that the PINN instance which was only trained on 50k epochs in Section 3.2.2 was still trapped at this local optimum which apparently conforms to the nonphysical prediction approaching the non-trivial fixed point (15). After approximately 90k epochs, however, the optimization manages to escape from this optimum and finally converges to the solution which is in good agreement with the true system dynamics. The minimum corresponding to this solution can be clearly seen in the upper right region (plus) of the loss landscape. For this example, the simulation length $T$ has to be reduced significantly, to a minimum of about $T = 0.01$, in order to observe the suboptimal location (diamond) completely vanishing from the loss landscape. This observation is in accordance with the domain decomposition method introduced in Mattey & Ghosh (2022) which divides the full time domain in 50 segments, thus using a step size of $\Delta t = 0.02$.

## 4 Discussion and Limitations

Our study and results suggest that fixed points of dynamical systems, irrespective of whether they are stable or unstable, lead to the formation of attractive optima in the physics loss landscape which influences the PINN training. This was demonstrated on a series of experiments using two simple dynamical systems described by ODEs, and two complex dynamical systems described by PDEs.

**On the Role of Fixed Points.** As discussed in Section 2.3, fixed points of dynamical systems are given by the roots of the nonlinear function $\mathcal{F}[u]$ in equation (1). Physics loss residuals (4) are thus small by definition since $\mathcal{F}[u] \approx 0$ and, thus, $u_t \approx 0$ in the vicinity of fixed points (see preparatory discussion in 3). These

properties seem to affect the training of PINNs: True fixed point solutions, i.e., solutions that are constant (ODEs) or steady-state (PDEs), trivially fulfill the physics loss function and, thus, these solutions yield local optima in the physics loss landscape. Those optima are attractive to the gradient descent optimization, as we could show in our experiments. Additionally, the fact that residuals are small in the vicinity of fixed points may make it more economical for the PINN to violate prescribed system dynamics locally, for the sake of settling at a "simpler" solution (Steger et al.). Effectively, fixed points, together with the $L_2$ losses in (7), allow PINNs to trade between severely violating the physics locally or approximating it inaccurately on the entire computational domain.

**Validity domain.** The question may arise as to which applications, in particular, the role of fixed points is critical in the training of PINNs. While, in general, this question is hard to answer due to the countless ways of studying dynamical systems, we can draw parallels between the systems studied in this work to provide an indication for similar expected behavior. First of all, our studied dynamical systems are time-continuous and described by – or at least can be brought into the form of – equation (1). Here we note that for any of our studied systems the differential operator $\mathcal{F}$ is not explicitly dependent on time $t$. This restricts our consideration to so-called autonomous differential equations. However, fixed points predominantly appear only in those specific systems, and many of the dynamical systems studied in the literature of PINNs and found in engineering problems are autonomous, such as fluid flow, diffusion processes or Hamiltonian systems. Whether or not a fixed point in a particular application will influence the PINN training is determined by several factors as concluded from our experiments. Our results indicate that having an IC near a fixed point or a trajectory that passes by a fixed point closely, combined with extended simulation times, can increase the chances of encountering training difficulties due to the fixed point. We further believe that in the context of Lyapunov stability the neighborhood of a fixed point, as determined by the differential operator $\mathcal{F}$, has an immediate effect on the influence of the fixed point in the PINN training.

**Remedies and Proposed Solutions.** In Section 1.1 we have already stated several remedies that have been proposed in order to overcome commonly-observed training difficulties in PINNs. The aim of this paper was not to propose a new entry to this list of remedies, but rather to give an explanation and further insights why some of them may work for the specific type of training difficulties faced on dynamical systems. Proposed solutions in this list range from techniques that reduce the computational domain, i.e., the simulation time in the context of dynamical systems, to methods that reweight or resample the dataset of collocation points. Our results showed that the simulation length $T$ has an immediate effect on the optimization landscape, and that smaller computational domains are often characterized by smoother landscapes in which undesired minima, corresponding to fixed points, are less pronounced or disappear altogether. We believe that several of the proposed collocation point sampling schemes affect the optimization landscape in a similar manner, effectively reducing the computational domain to regions in which collocation points are sampled densely (e.g., close to the boundary of the computational domain, as in Daw et al. (2022) or Wang et al. (2022)). Finally, as we have argued in Section 3, only the provision of IC/BC data can prevent a PINN getting stuck in an undesired fixed point solution. Even so, the fixed point may still be attractive, leading either to small violations of the physics loss (as shown on the toy example in Section 3.1.2) or to small violations of the IC. This may partly explain the success of loss weighting schemes, and the fact that sometimes greater weights on the IC/BC loss are required to arrive at the desired solution (Wang et al., 2021a; Maddu et al., 2021; Jin et al., 2021).

**Limitations and Further Work.** One may argue that the trade-off inherent in PINNs, or the multi-objective nature of their vanilla formulation (7), should be vacuous, as the true solution satisfies both physics and the IC/BC. Thus, nonphysical predictions should never represent a better optimum than the desired, physical solution. This is true, however, only for PINNs with unlimited expressive power. In practice, the expressivity of a neural network is always limited by its (necessarily) finite size. Therefore, the mentioned trade-off is effective, and we have reason to believe that there are settings where the desired solution does not correspond to a global optimum (cf. Figure 1). Future work shall investigate this aspect from a more theoretical perspective, instantiating approximation theorems for neural networks for PINNs.

Further, one may argue that some of the observed nonphysical predictions simply appear because the PINN has not sufficiently converged. In other words, training was not long enough to depart from the (flat) IC, which may correspond to a trivial solution of the differential equation (Wong et al., 2021; Leiteritz

& Pflüger, 2021). A large part of the literature on propagation failures (see Section 1.1.2) points in this direction. Further, such a statement is supported by the fact that the physics loss of, e.g., the solution approaching the stable fixed point in Figure 6 is high, and by the late transitions to the correct solutions in Section 3.3. Indeed, we do not claim that minima of the physics loss formed by the presence of fixed points *always* correspond to (good) minima of the full loss (7) – these minima may disappear entirely (e.g., for small computational domains), turn into saddle points that slow down convergence, or achieve a wider basin of attraction and/or smaller loss than the minimum corresponding to the true solution, in the extreme case where ICs are very close to unstable fixed points.

Finally, our investigations are based on a selection of dynamical systems. We chose these systems because they are intuitive to understand, yet still exhibit nontrivial dynamics. Moreover, the particular choice of these systems allowed us to separate the effect of different types of fixed points and to at least partly exclude other explanations for the training difficulties of PINNs. Future work shall be devoted to studying fixed points and steady-state solutions, and to the wider spectrum of asymptotic properties of solutions to dynamical systems.

## 5    Conclusion

In this paper, we studied the physics loss optimization in PINNs when applied to dynamical systems governed by differential equations. Our results revealed that nonphysical predictions appear as attractive optima in the physics loss landscape and seem to stem from the presence of fixed points inherent to dynamical systems. These minima or saddle points potentially disrupt and trap the gradient descent optimization, leading to commonly observed convergence issues in PINNs. Reducing the computational domain yielded a greater rate of training success and, in general, reduced the complexity of the physics loss optimization. In the future, we believe that interdisciplinary research that includes advances in deep learning, stability theory and/or a further understanding of the underlying physics may improve physics-informed machine learning or benefit from it.

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

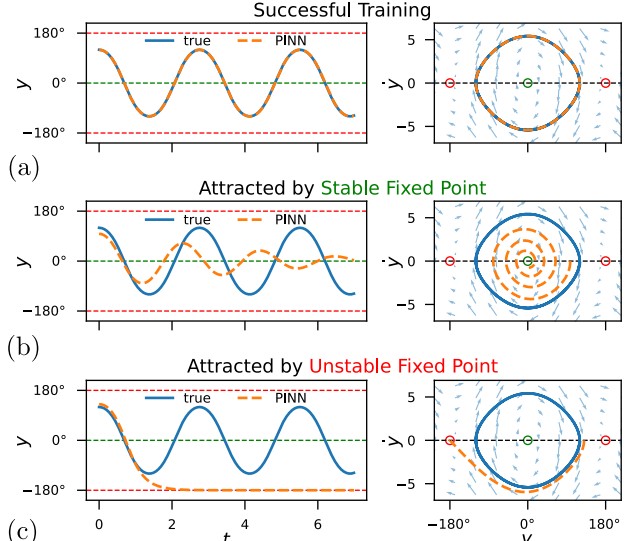

Figure 6: **Undamped Pendulum.** Representative examples of training cases from Table 1. Training outcomes are classified as either (*a*) successful training, (*b*) attracted by stable fixed point, or (*c*) unstable fixed point. *Left*: predicted trajectories as function of physical time. *Right*: predicted trajectories in phase space show whether their end positions $y$ and $\dot{y}$ lie on, inside or outside the orbit of the true pendulum dynamics. Green and red lines/circles represent stable and unstable fixed points, respectively.

## A    Software and Hardware Specifications

All code in our experiments is implemented in Python version 3.8 and TensorFlow version 2.9.1. Computations are performed on a Nvidia Tesla T4 GPU with a memory size of 16 GB.

## B    Additional Content to Undamped Pendulum

This section provides further results in addition to the in Section 3.1.1 and 3.1.2 presented experiments.

### B.1    Classification of Training Outcomes in Phase Space

In the main part of this paper, we classify unsuccessful training outcomes to be either influenced by the stable ($y^* = 0°$) or the unstable fixed point ($y^* = 180°$). This classification is conducted by determining the trajectories' end position in phase space, i.e., by evaluating $y$ and $\dot{y}$ (see Figure 6). Based on this, end positions that lie inside the periodic orbit of the true pendulum dynamics are classified as being attracted by the stable fixed point, whereas positions that lie outside the orbit are considered as being attracted by the unstable fixed point. Here we note that trajectories of unsuccessful training outcomes by default leave the periodic orbit which renders this separation possible.

### B.2    Rate of Training Success - Optimization Settings

In Table 1, all PINN instances are optimized with an initial learning rate $\alpha = 0.001$, number of collocation points $N_c = 64$, loss weighting factor $\lambda = 1$ and as network initialization the Glorot uniform initializer. In addition to this, we also test different optimization settings using the 4x50 network architecture with tanh activation as base model.

Table 2: **Undamped Pendulum.** Rate of training success across different system and optimization settings using the 4x50 network architecture with tanh activation. Triplets in the main table represent in percentage (%) and in the respective order, cases of successful training, attracted by stable fixed point, and unstable fixed point. The tested optimization settings are learning rate ($\alpha$), number of collocation points ($N_c$), loss weighting factor ($\lambda$) and network weights initialization (*Init.*) with He denoting the *He uniform* initialization. Baseline model (see Table 1) uses $\alpha = 0.001$, $N_c = 64$, $\lambda = 1$ and Glorot uniform initialization. Bold triplets represent a low ($< 5\%$) success rate.

| $T$ | | 2.5 | | | 5 | | | 7.5 | | |
|---|---|---|---|---|---|---|---|---|---|---|
| $y_0$ | | 25° | 100° | 175° | 25° | 100° | 175° | 25° | 100° | 175° |
| Baseline | | 98/2/0 | 100/0/0 | 100/0/0 | **0/100/0** | 90/10/0 | **0/100/0** | **0/100/0** | **0/100/0** | **0/61/39** |
| $\alpha$ | 0.01 | 37/0/63 | 36/0/64 | 60/0/40 | 16/0/84 | 19/0/81 | 7/0/93 | **0/7/93** | 13/0/87 | **0/0/100** |
| | 0.001 | **0/100/0** | **0/100/0** | **0/100/0** | **0/100/0** | **0/100/0** | **0/100/0** | **0/100/0** | **0/100/0** | **0/100/0** |
| $N_c$ | 16 | 96/4/0 | 100/0/0 | 100/0/0 | **0/100/0** | **4/96/0** | **0/100/0** | **0/100/0** | **0/100/0** | **0/100/0** |
| | 256 | 100/0/0 | 100/0/0 | 100/0/0 | **0/100/0** | 100/0/0 | 35/64/1 | **0/99/1** | **0/100/0** | **0/89/11** |
| $\lambda$ | 0.1 | **0/100/0** | 61/39/0 | 93/7/0 | **0/100/0** | **0/99/1** | **0/85/0** | **0/100/0** | **0/100/0** | **0/100/0** |
| | 10 | 100/0/0 | 100/0/0 | 100/0/0 | 11/89/0 | 42/0/0 | 6/85/9 | **0/99/1** | **0/100/0** | **0/35/65** |
| *Init.* | He | 100/0/0 | 98/0/2 | 100/0/0 | **0/98/2** | 93/7/0 | **0/98/2** | **0/98/2** | **0/98/2** | **0/73/27** |

Results to this experiment can be found in Table 2, where we included the baseline model with default optimization settings from Table 1 as reference. In general, we observe that none of the tested optimization settings yields substantial improvement for IC close to fixed points and long simulation times.

### B.3  Rate of Training Success - Further Thresholds

In the main part of the paper, we use for the classification of successful training a threshold of 15% in terms of the $L_2$ relative error. To demonstrate that our particular choice of this threshold does not contradict qualitative conclusions made in the main part, we further provide results using different thresholds. In particular, we validate the in Table 1 presented training cases now with a threshold of 5% and 25% in terms of the $L_2$ relative error.

The results can be found in Table 3. For compactness, we only show the results for the 4x50 and 8x100 architecture with tanh activation. As apparent in the table, no substantial differences in the classified training outcomes can be observed when using different thresholds.

### B.4  Fixed Points Becoming Economical Solutions

In the main part of this work, Figure 1 shows for the toy example that nonphysical predictions can become better minima than that of the desired solution when the IC is close to a fixed point. To demonstrate similar qualitative observations for the undamped pendulum, we perform an experiment similar to that in Section 3.1.2.

In particular, we use a 8x100 network architecture with tanh activation and choose a simulation time of $T = 10$. Since for this simulation time physics-driven PINN instances will hardly converge to the true solution, we implement the same data-guided strategy as presented in Section 3.1.2: We include a total number of 100 labeled training points in the first half of the training. The data is sampled from the Runge-Kutta solution, equidistantly in the computational domain. In the second half, the training continues with the physics loss optimization only. Indeed, as show in Figure 7(*b*), the data-guided strategy successfully converges to the true solution, representing successful outcomes and sufficient expressive power for the chosen PINN architecture. We again compare this approach to physics-driven training, i.e., without the use of labeled training data. We repeat for each IC the experiment with 10 uniquely initialized PINN instances per training strategy. We note that none of the physics-driven instances converges to the true solution (see Figure 7(*b*)). The unsuccessful training outcomes are further classified into whether the subsequent nonphysical prediction violates the physics by either being attracted by the stable or the unstable fixed point (see Figure 6).

Table 3: **Undamped Pendulum.** Rate of training success using differently set thresholds in terms of the $L_2$ relative error for the classification of successful and unsuccessful training. Triplets in the main table represent in percentage (%) and in the respective order, cases of successful training, attracted by stable fixed point, and unstable fixed point. Bold triplets represent a low ($< 5\%$) success rate.

| | $T$ | 2.5 | | | 5 | | | 7.5 | | |
|---|---|---|---|---|---|---|---|---|---|---|
| | $y_0$ | $25°$ | $100°$ | $175°$ | $25°$ | $100°$ | $175°$ | $25°$ | $100°$ | $175°$ |
| | $L_2 < 5\%$ | 98/2/0 | 100/0/0 | 100/0/0 | **0/100/0** | 90/10/0 | **0/100/0** | **0/100/0** | **0/100/0** | **0/61/39** |
| 4x50 | $L_2 < 15\%$ | 98/2/0 | 100/0/0 | 100/0/0 | **0/100/0** | 90/10/0 | **0/100/0** | **0/100/0** | **0/100/0** | **0/61/39** |
| | $L_2 < 25\%$ | 99/1/0 | 100/0/0 | 100/0/0 | **0/100/0** | 90/10/0 | 34/66/0 | **0/100/0** | **0/100/0** | **0/61/39** |
| | $L_2 < 5\%$ | 43/37/20 | 84/0/16 | **0/29/71** | 100/0/0 | 97/0/3 | 43/49/8 | 43/37/20 | 84/0/16 | **0/29/71** |
| 8x100 | $L_2 < 15\%$ | 100/0/0 | 100/0/0 | 99/0/1 | 100/0/0 | 97/0/3 | 92/0/8 | 49/31/20 | 84/0/16 | **1/29/70** |
| | $L_2 < 25\%$ | 100/0/0 | 100/0/0 | 99/0/1 | 100/0/0 | 97/0/3 | 92/0/8 | 55/25/20 | 84/0/16 | **1/29/70** |

In Figure 7(a) we report the minimal physics loss values across all epochs for each training outcome. We observe, similar to the behavior in the toy example, that the PINN predictions attracted by the unstable fixed point ($y^* = 180°$) achieve lower physics losses as the IC gets closer to it. Furthermore, for $y_0 = 175°$ the nonphysical solution becomes a better optimum than that of the desired solution. As a direct consequence, any physics-driven instance converges to it.

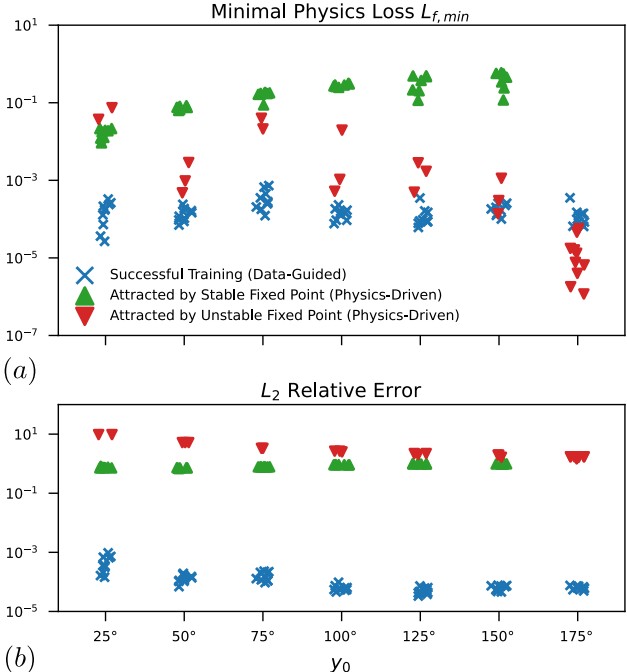

Figure 7: **Undamped Pendulum.** (a) Minimal physics loss across all epochs. (b) $L_2$ relative error. While the data-guided PINNs converge to the true solution (blue), the physics-driven PINNs yield incorrect system dynamics (large $L_2$ relative errors) by being either attracted by the stable (green) or unstable (red) fixed point (see Figure 6). For $y_0 = 175°$ the nonphysical solution becomes a better optimum than that of the desired solution. In the figure, markers were randomly shifted horizontally to reduce overlap.

| $T$ | | 2.5 | | | 5 | | | 7.5 | | |
|---|---|---|---|---|---|---|---|---|---|---|
| $y_0$ | | 0.001 | 0.01 | 0.1 | 0.001 | 0.01 | 0.1 | 0.001 | 0.001 | 0.1 |
| | tanh | 100 | 100 | 100 | **1** | **1** | 12 | **1** | **2** | 14 |
| 4x50 | swish | 100 | 100 | 100 | 69 | 72 | 100 | **0** | **0** | **2** |
| | sin | 100 | 100 | 100 | 13 | 5 | 61 | **3** | **4** | 12 |
| | tanh | 100 | 100 | 100 | **0** | **0** | **3** | **0** | **1** | 10 |
| 8x100 | swish | 100 | 100 | 100 | 91 | 100 | 100 | **0** | **0** | **0** |
| | sin | 100 | 100 | 100 | **0** | **1** | 13 | **0** | **3** | 14 |

Table 4: **Toy Example.** Rate of training success across different system settings ($T$ and $y_0$) and network architectures (size and activation function). Numbers in the main table represent in percentage (%) cases of successful training. Bold numbers represent a low ($< 5\%$) success rate.

## C  Additional Content to Toy Example

### C.1  Analytical Solution

The analytical solution to (10) is given by

$$
y(t) = \begin{cases} \left(1 + \left(\frac{1}{y_0^2} - 1\right) e^{-2t}\right)^{-1/2} & \text{for } 1 \geq y_0 > 0, \\ 0 & \text{for } y_0 = 0, \\ -\left(1 + \left(\frac{1}{y_0^2} - 1\right) e^{-2t}\right)^{-1/2} & \text{for } 0 > y_0 \geq -1. \end{cases}
$$

### C.2  Rate of Training Success

As defined in Section 3.1.1, we declare training successful if the $L_2$ relative error is below 15%. We show the rate of training success for the toy example using different network architectures in Table 4. Similar to observations on the undamped pendulum (see Table 1), we observe a low rate of training success for IC close to the unstable fixed point ($y^* = 0$) and long simulation times.

## D  Vortex Shedding

In this section we provide additional information to the experimental setup for simulating vortex shedding, including the computational domain, boundary conditions, as well as network and optimization settings. A publicly available database of direct numerical simulation data can be found in Boudina (2021).

### D.1  Computational Domain and Boundary Conditions

The computational domain for this experiment is set to the compact space $(x, y) \in \Omega := [-5, 15] \times [-10, 10]$, where a cylindrical body is placed at $(x, y) = (0, 0)$ with a diameter of $d = 1$. For our main experiment, the top/bottom boundary is considered as a moving wall with no-slip conditions, where for the outlet zero-gradient conditions are applied. The cylinder boundary is chosen as no-penetration and no-slip condition. These BC are also shown in Figure 8.

Here we note that we have also performed experiments with a different computational domain and different BCs (top/bottom boundary with symmetry wall and zero-gradient, and outlet with zero pressure), but none of the tested settings led to the desired vortex shedding motion, but showed similar qualitative behavior as presented in the main part of this work.

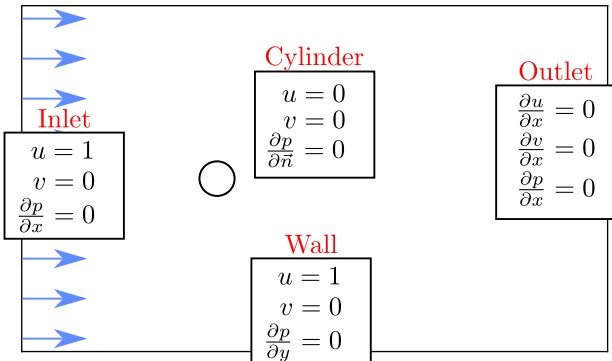

Figure 8: **Fluid Dynamics.** Boundary conditions for vortex shedding.

## D.2 Optimization and Data Settings

The overall loss function in this experiment is composed of the respective loss functions for the BCs, the physical and additional data constraints:

$$L(\theta) = L_{\text{Inlet}}(\theta) + L_{\text{Outlet}}(\theta) + L_{\text{Wall}}(\theta) + L_{\text{Cylinder}}(\theta) + L_{f,x}(\theta) + L_{f,y}(\theta) + L_{Data}(\theta), \tag{17}$$

where $L_{f,x}$ and $L_{f,y}$ are the physics loss functions for (12a) and (12b), respectively. Here, $L_{Data}$ imposes the (initial) sequence of the developed vortex shedding flow field, which for the physics-driven PINN is 50% of the first period, and for the data-guided PINN three consecutive periods.

The data set for this initial sequence is sampled from the reference database with a batch size of $N_{IC,\text{batch}} = 1024$. At each batch iteration, collocation points and training data for the BC are sampled anew with sizes $N_{col} = 1024$, $N_{\text{Inlet}} = 128$, $N_{\text{Outlet}} = 128$, $N_{\text{Wall}} = 256$ and $N_{\text{Cylinder}} = 128$.

As already stated in the main part of this work, we introduce a stream function $\psi(t, \vec{x})$ with $u = \psi_y$ and $v = -\psi_x$ to enforce continuity, i.e., conservation of mass for an incompressible fluid. A single 8x100 neural network with tanh activation functions is then used to approximate $\psi_\theta(t, \vec{x})$ and $p_\theta(t, \vec{x})$.

Different optimization settings have been tested throughout our work, but none of them led to the desired vortex shedding for the physics-driven PINN. For the sake of simplicity, we only state the settings used for the experiment presented in the main part of this paper: Optimization is performed using Adam with default settings for the moment estimates and a total number of 10k epochs. The initial learning rate is set to $\alpha = 0.001$ and an exponential decay with rate 0.9 and step 1000 is applied. Training for this setting and for the in Section A listed software/hardware took about $7h$.

## E  Proof to Allen-Cahn Equation

*Proof.* For the proof, note that for (15), we have that $u_t \equiv 0$. We hence remain to investigate the right-hand side of (14). To this end, note that $u(x, t)$ is piecewise constant. Specifically, we have that $u_x(x') = 0$ for $x' \in [-1, 1] \setminus \{-0.5, 0.5\}$, hence also $u_{xx}(x') = 0$. Since further, for all $x \in [-1, 1]$ we have that $u^3(x, t) = u(x, t)$, the right-hand side of (14) is zero on $x' \in [-1, 1] \setminus \{-0.5, 0.5\}$.

Suppose now that the collocation points on which $L_f(\theta)$ is evaluated are drawn from a continuous distribution supported on $[-1, 1] \times [0, T]$, i.e., from a distribution that is absolutely continuous w.r.t. the Lebesgue measure on $\mathbb{R}^2$. Now, the subset $\{-0.5, 0.5\} \times [0, T]$ is a Lebesgue null set of $\mathbb{R}^2$, hence the probability that collocation points are drawn from this set is zero. Since the physics loss is evidently zero outside of this set, this completes the proof. $\square$

## F    Visualizing the Physics Loss Landscape

Visualization of the loss landscape is based on the work of Li et al. (2018) and is a two-dimensional projection. We plot the loss landscape $L(\theta_1, \theta_2)$ with two specific directions $\theta_1$ and $\theta_2$ as stated in the main part of this work. A basic Gram-Schmidt process is then used to obtain an orthonormalized set of the two directions. The loss landscape for different simulation times $T$ is obtained by evaluating the physics loss function on a total number of 1024 collocation points, sampled from the time domain $t \in [0, T]$.

For the toy example, loss values greater than $L(\theta_1, \theta_2) > 0.2$ were truncated to highlight the interesting domain in Figure 4(c)-(f). Similarly, loss values greater than $L(\theta_1, \theta_2) > 10$ were truncated for the Allen-Cahn example where we also use a logarithmic presentation of the loss landscape to highlight the interesting domain in Figure 5(d)-(g).

