# OpenReview forum: "On the Role of Fixed Points of Dynamical Systems in Training Physics-Informed Neural Networks"
_TMLR — Accepted by TMLR_

### Review · Reviewer_djNt · 2022-11-09

**Summary Of Contributions:**

The paper adresses a typical problem in PINNs: the existance of bad local minima in the loss functions, which correspond to overly simple solutions. The paper makes an empirical study using previously existing methods to collect evidence that these problems are connected to fixed points in the dynamical system. Section 3 show that the relevant problems exist for PDEs. Section 4 includes demonstrations on some examples, where fixed points probably induces bad learning behavior. The loss landscape on a toy example demonstrates that the proposed effect increases over long times, and hence makes PINN (as they are currently being used) unsuitable for learning of dynamical systems over long time periods.

**Audience:**

Yes

**Broader Impact Concerns:**

None.

**Claims And Evidence:**

Yes

**Requested Changes:**

 - Section 2:
   - Not every dynamical system is of the form given in (1), and not even every dynamical system can be described by differential equations. Please be honest and state that you are considering dynmical systems as in (1), which is certainly an imporant class. (For example, the system in (10) is not of this form, although the system (10) can of course easily be modified to satisfy (1).)
   - Please state necessary assumption on $\Omega$, e.g. bounded, compact, simply connected, regularity assumptions on its boundary, ...
   - Is your boundary operator necessarily linear?
   - After Formula (5): you probably mean $\times$ instead of $\otimes$. (See also Appendix B)
 - Section 3:
   - Please clearly state the assumptions made on the solutions of the NS equations, in particular on (in)compressibility, in the view of using the stream function.
 - Section 4:
   - Plots do not really have colorblind people in mind. And they do not really work in grayscale.

This is more a suggestion than a requirement: the literature on PINNs is well-cited. One might (but not must) cite alternative approaches to differnetial equations using ML, e.g. (amonst many) the following arxiv links: 1406.2582, 1703.00787, 1801.09197, 2103.10153, 2103.12959, 2110.11812, 2202.01287, 2205.03185, 2208.01565, 2208.12515

**Strengths And Weaknesses:**

Strengths:
 - The paper shows clear experimental evidence with good experimental setups, that the proposed problem in training PINNs has actually somethin to do with fixed points.
 - The experiments are support by physical and mathematical explanations, which ar eno proofs, but at least intuitively point to the empirical findings.

Weaknesses
 - No proper solution to the problem is given, although some weakly mitigating effects are being demonstrated
 - Rather few test cases have been done, and mostly on ODEs

---

> ### Author Response · Authors · 2022-11-30
> **Response to Reviewer djNt**
>
> We thank the reviewer for the valuable suggestions. All changes are color-coded in our revised manuscript.
>
> ### No Solutions to the Problem are Given
> The purpose of our paper is not to solve the long-standing problem of successful PINN training; there already exist many proposed remedies, such as domain decomposition, loss weighting, or adaptive sampling of collocation points (see our Introduction). Our paper essentially explains why these remedies for training problems work. To make that clearer, we have adapted our discussion section accordingly.
>
> ### Focus on ODEs
> We understand that the two toy examples, which are simple ODEs, may not be convincing, especially since the Navier-Stokes example was listed only for motivation. To this end, we not only restructured our paper, making the Navier-Stokes example part of the main storyline, but we also added an additional PDE, the one-dimensional Allen-Cahn equation. This results in four settings that cover a variety of properties:
> - transient simulation of a toy ODE with two stable and one unstable fixed points, solved by a hard-constrained PINN
> - periodic behavior of a pendulum affected by both stable and unstable fixed points, solved by a vanilla PINN
> - periodic behavior of the Navier-Stokes equation (vortex shedding) affected by a stable fixed point
> - transient simulation of the Allen-Cahn equation, with a non-trivial fixed point, solved by a PINN with a high weight on the IC loss
>
> ### Further Changes
> We have performed all requested changes.:
> - We have added a clarification that the second-order ODE of the pendulum can be converted to a coupled system of two first-order ODEs (after the system is introduced in equation 9).
> - We now clearly state the assumption on $\Omega$ and the boundary operator $\mathcal{B}$ (Section 2.1.).
> - We changed $\otimes$ to $\times$ when stating the computational domains.
> - We now state the assumption on incompressibility of the fluid flow (equation 13).
> - We have updated the plots to be readable for colorblind people and to also work in greyscale.
> - We have further surveyed the suggested references and have concluded that they are largely out of this work’s scope. However, we introduced the occasional citation in the introduction helping the reader to situate our work in the wider field.

---

### Review · Reviewer_jz2A · 2022-11-11

**Summary Of Contributions:**

This paper examines numerically the inconsistent training outcomes of physics informed neural networks (PINNs) in some simple systems after motivating the problem by discussing an unsuccessful application to the Navier-Stokes equation. The authors hypothesize that a lack of convergence arises due to fixed points in the dynamics of the target dynamical system. The effect of fixed points is studied from the perspective of the loss landscape and optimization dynamics for 1) learning the dynamics of an undamped pendulum and 2) a gradient system with two stable fixed points and an unstable fixed point.

**Audience:**

Yes

**Broader Impact Concerns:**

No, none come to mind.

**Claims And Evidence:**

No

**Requested Changes:**

I think in order for me to recommend this study for publication, there are several additional things I would need to see:

1. There needs to be some actionable proscription. If a PINN is not converging well because of a fixed point, how should one simulate so as to improve training.

2. There must be a better plausibility argument for why training fails near fixed points. The argument that stable fixed points lead to slow training dynamics could almost certainly be made more precise and illustrated in a more general setting.

3. On a related note, some more foundational understanding of this phenomenon would be helpful to connect these observations to nontrivial systems.

**Strengths And Weaknesses:**

Strengths:

- The paper formulates a clear hypothesis to diagnose the challenges in training PINNs.
- PINNs are a timely and important topic in the machine learning literature, especially for applied mathematics and computational sciences.

Weaknesses:

- This study does not engage in any real hypothesis testing, rather it simply provides two minimal examples where fixed points influence the convergence of learning. Is it always the case? What about at the bottom of a quartic potential, where there's a stable fixed point with large fluctuations?

- The evidence is purely numerical and is guided by models that are very simplistic. It's hard to connect the results obtained in the paper to the Navier Stokes solution presented at the beginning.

- There is not a compelling explanation to demonstrate that this phenomenon is generic. The argument appears to be by analogy.

- The connection to the Navier-Stokes example isn't very clear. The dynamics resembles an unstable fixed point, but is it?

---

> ### Author Response · Authors · 2022-11-30
> **Response to Reviewer jz2A**
>
> We thank the reviewer for the constructive feedback. All changes are color-coded in our revised manuscript.
>
> ### Missing Hypothesis Test, Purely Numerical Evidence, Foundational Understanding Why Training Near Fixed Points Fails
> While many important results in the field of understanding deep learning are exclusively experimental (at least at first), we agree that a more fundamental coverage of the topic is desirable. Indeed, the understanding of gradient-based training of neural networks improves steadily, both by experimental and theoretical evidence. In that regard, however, we see our paper as one that proposes a hypothesis and supports it by anecdotal experimental evidence, rather than as a paper that tests a hypothesis using a rigorous statistical methodology. We believe that such an in-depth analysis of the phenomenon is beyond the scope of this study and, judging by the page limitations, of the considered venue.
>
> However, we argue that we have obtained and provide to the readers a foundational understanding of why fixed points are attractive for gradient-based optimization. Already our initial submission argued in the discussion that fixed points correspond to weight configurations for which the physics loss vanishes. The reviewer comment made us aware that the presentation of this insight was insufficient, which is why we have expanded on this point now at the beginning of Section 3. Further, for the newly added Allen-Cahn equation, we have provided a proposition (Proposition 1) that characterizes a non-trivial fixed point of this system, at least when it is implemented using PINNs.
>
> ### Connection to Navier-Stokes Is Not Clear
> We agree it takes a leap of faith to carry over conclusions drawn for two simple ODEs to more complicated PDEs such as the Navier-Stokes equation. First, note that we do not claim that the PINN solution in Figure 1 (old manuscript) is a fixed point of the Navier-Stokes equation, but rather that this solution is obtained because gradient-based optimization is affected by such a fixed point. To support this claim, we have restructured Figure 2 (revised manuscript) to now show the time evolution of $F_x$ and $F_y$, which indeed approach zero for larger values of $t$ (indicating fixed point properties). Second, we have added the Allen-Cahn equation, which is a PDE with a non-trivial fixed point (Proposition 1).
>
> ### Actionable Proscriptions
> The purpose of our paper is not to provide actionable proscriptions; there already exist many, such as domain decomposition, loss weighting, or adaptive sampling of collocation points (see our Introduction). Our paper essentially explains why these remedies for training problems work. To make that clearer, we have adapted our discussion section accordingly.

---

### Review · Reviewer_KDWh · 2022-11-16

**Summary Of Contributions:**

The paper suggests, through a series of experiments over two toy examples and one complex dynamical system, Navier-Stokes equation, that fixed points shape local optima of the loss landscape. This would help inform the architecture and training of the model. The authors provide empirical evidence to support this claim.

**Audience:**

Yes

**Broader Impact Concerns:**

No broader impact concerns registered

**Claims And Evidence:**

No

**Requested Changes:**

- Figure 1: it is important to clarify why cannot the given performance be just explained by the previously documented phenomenon when trained neural networks are giving good interpolations but bad extrapolations (see Ziyin et al, 2020; Davini et al, 2021)?
- It is important to show the phenomenon for multiple complex dynamical systems. This may include varying the parameterisation of Navier-Stokes equation such as varying Reynolds number, as well as other dynamic systems (maybe Lorenz system?). The main concern, otherwise, is that while there is evidence on two particular toy problems, the evidence of generalisation to the more complex problems such as solving Navier-Stokes equation is not entirely conclusive.
-In Table 1, although the general meaning is clear, the reviewer struggled to find the exact procedure for evaluation of the following values: 'attracted by stable fixed point, and unstable fixed point'. It is important that the authors clarify upon it.
- Figure 4 gives good qualitative motivation why would the fixed point affect optimisation landscape; however, is it possible to conclusively clarify this for multiple tasks by showing landscapes for different tasks?

Davini, David, Bhargav Samineni, Benjamin Thomas, Amelia Huong Tran, Cherlin Zhu, Kyung Ha, Ganesh Dasika, and Laurent White. "Using physics-informed regularization to improve extrapolation capabilities of neural networks." (2021)
Ziyin, Liu, Tilman Hartwig, and Masahito Ueda. "Neural networks fail to learn periodic functions and how to fix it." Advances in Neural Information Processing Systems 33 (2020): 1583-1594.

**Strengths And Weaknesses:**

Strengths:
- the research question above sounds like a novel aspect of training physically-informed machine learning methods, and it is well grounded in the literature  as described by the authors in the introduction
- the results are underpinned by the experiments on different models and illustrate the given phenomenon

Weaknesses:
- the main concern is about the sufficiency of evidence: the generalised conclusions of impact of fixed points  on the optimisation of physics-informed neural networks only rely upon a limited number of experiments; therefore the evidence for the claims should be improved, and that is the reason behind the answer on the question "Are the claims made in the submission supported by accurate, convincing and clear evidence? " is currently No (see comments in the requested changes section)

---

> ### Author Response · Authors · 2022-11-30
> **Response to Reviewer KDWh**
>
> We thank the reviewer for the insightful comments. All changes are color-coded in our revised manuscript.
>
> ### Insufficient Evidence
> We understand that the two toy examples, which are simple ODEs, may not be convincing, especially since the Navier-Stokes example was listed only for motivation. To this end, we not only restructured our paper, making the Navier-Stokes example part of the main storyline, but we also added an additional PDE, the one-dimensional Allen-Cahn equation.
> This results in four settings that cover a variety of properties:
> - transient simulation of a toy ODE with two stable and one unstable fixed points, solved by a hard-constrained PINN
> - periodic behavior of a pendulum affected by both stable and unstable fixed points, solved by a vanilla PINN
> - periodic behavior of the Navier-Stokes equation (vortex shedding) affected by an unstable fixed point
> - transient simulation of the Allen-Cahn equation, with a non-trivial fixed point, solved by a PINN with a high weight on the IC loss
>
> We believe that, as a whole, these four examples illustrate that PINN training is affected by fixed points and that, thus, our conclusions are supported by sufficient evidence.
>
> ### Interpolation vs. Extrapolation
> Indeed, Fig. 1 in our previous manuscript suggested this interpretation. However, note that PINNs, when used for simulating dynamical systems, are essentially always extrapolating. This is particularly apparent in initial value problems, where the solution to the ODE/PDE is given only at $t_0=0$ (and maybe at the boundary of the computational domain), and resonates with Davini et al., who used the physics loss to improve extrapolation. Therefore, both interpolation (between the boundaries) and extrapolation (from $t=0$ to later times) can be accomplished by minimizing the residual at the collocation points. The general concept of inter- vs. extrapolation does therefore not apply to PINNs in the same way as it does for standard neural networks.
>
> ### Fixed Points Affect Optimization Landscape
> To show that the optimization landscape is affected by fixed points also for different tasks, we have added corresponding plots for the Allen-Cahn equation.
>
> ### Exact Procedure for Evaluating Training Outcomes on Pendulum
> We now have provided further details on the exact procedure for evaluating the training outcomes for the pendulum system in Appendix B.1 and placed references in the main part accordingly.
>
> ### Further Comment on Varying Reynolds Number
> Although studying a different parameterization of the Navier-Stokes by varying the Reynolds number would have been delightful, we were limited by the availability of reliable reference data. Moreover, as now made clear in the revised work, the periodic dynamics of vortex shedding only appears above a critical Reynolds number $\text{Re}_{crit}\approx48$, rendering our particular choice of $\text{Re}=100$ optimal for avoiding the consideration of turbulent vortex shedding at higher Reynolds numbers. We have tested different parametrizations in terms of boundary conditions and the computational domain. None of the tested settings, however, showed a different outcome as presented in the paper.

---

### Review · Reviewer_J9px · 2022-11-25

**Summary Of Contributions:**

This paper investigates the limitations of physics-informed neural networks (PINNs).
They show that (stable and **unstable**) fixed points in the PDE contribute to the optimization difficulties of PINN since they are attractive (in terms of loss minimization) for the PINN, thus leading the PINN to learn non-physical solutions.

**Audience:**

Yes

**Broader Impact Concerns:**

No concerns

**Claims And Evidence:**

No

**Requested Changes:**

- I would like to see experiments with larger/deeper NNs to see if this problem remains there.
- Less toyish tasks.

Minor: I would clarify the notation $u_{\theta,t}$

**Strengths And Weaknesses:**

## strength:
The message of this paper is quite simple: PINNs may get stuck in some local minima that correspond to fixed points of the dynamics.

## weaknesses:
- The experiments are very toyish (small neural networks and easy tasks). It is unclear if larger/deeper neural networks would achieve the same conclusion.
- I found the notation $u_t$ for the derivative quite confusing since it is eventually mixed with $u_\theta$ and $u_{\theta,t}$ where the subscript $\theta$ does not correspond to any derivation.
- The paper does not investigate/propose solutions on how to fix that issue. However, I acknowledge that this may be beyond the scope of this paper.


## summary

Overall I feel like since this paper is an experimental paper, the evidence proposed by the author is relatively weak (toy experiments and small NNs) So, I feel that this paper is currently slightly under the acceptance threshold. Stronger experiments would be required to strengthen the claims.

---

> ### Author Response · Authors · 2022-11-30
> **Response to Reviewer J9px**
>
> We thank the reviewer for the constructive feedback. All changes are color-coded in our revised manuscript.
>
> ### Simple Examples
> We understand that the two toy examples, which are simple ODEs, may not be convincing, especially since the Navier-Stokes example was listed only for motivation. To this end, we not only restructured our paper, making the Navier-Stokes example part of the main storyline, but we also added an additional PDE, the one-dimensional Allen-Cahn equation. This results in four settings that cover a variety of properties:
> - transient simulation of a toy ODE with two stable and one unstable fixed points, solved by a hard-constrained PINN
> - periodic behavior of a pendulum affected by both stable and unstable fixed points, solved by a vanilla PINN
> - periodic behavior of the Navier-Stokes equation (vortex shedding) affected by a stable fixed point
> - transient simulation of the Allen-Cahn equation, with a non-trivial fixed point, solved by a PINN with a high weight on the IC loss
>
> ### Effect of Network Architecture
> Indeed, it has been observed that the architecture (layer width, layer number) of PINNs affect their performance. We have covered several architectural options in our sensitivity analysis (Tables 1 and 4), showing that the general qualitative picture remains the same. Further, for several of our examples, we have also trained standard data-driven PINNs, i.e., by supporting the PINN with ground truth data. These experiments confirmed that the considered network architectures have sufficient expressive power to cover the complexity of the desired solutions.
>
> ### No Solutions to the Problem are Given
> We agree that an in-depth discussion of solutions is beyond the scope of the paper; there already exist many proposed remedies, such as domain decomposition, loss weighting, or adaptive sampling of collocation points (see our Introduction). Our paper essentially explains why these remedies for training problems work. To make that clearer, we have adapted our discussion section accordingly
>
> ### Further Changes
> We have added a clarification of the notation $u_{\theta,t}$ after it is introduced in equation 4.

---

### Author Response · Authors · 2022-11-30
**General Response to Reviewers**

We thank the reviewers for their constructive feedback and their valuable time. We have taken all comments into consideration and have revised our manuscript accordingly. Specifically, we have

- added experiments with the Allen-Cahn equation, a one-dimensional PDE often used as a benchmark in the PINN literature and
- showed analytically that it has a non-trivial fixed point that seems to be attractive for PINN training;
- restructured our paper accordingly, placing less emphasis on the two simple ODE example and improved our presentation of the two discussed PDEs (Navier-Stokes and Allen-Cahn);
- revised all parts of the paper to improve their readability.

All changes are highlighted in blue in our revised manuscript.
We further respond to all reviewers' concern separately in the corresponding responses.

---

### Decision · Action_Editors · 2023-01-07

**Recommendation:** Accept with minor revision

**Comment:**

This paper analyses the training dynamics of physically-informed neural networks (PINNs), especially to study their failure cases.
The main hypothesis in this work is that fixed points in differential equations can negatively impact the loss landscape when training PINNs, leading to nonphysical predictions. The authors conjecture that these local minima are responsible of the convergence issues usually observed with PINNs, especially for initial conditions close to fixed points or for long forecasting horizons.

The submission initially received mixed reviews, with two slightly positive and two slightly negative recommendations. The main concerns pointed out by reviewers related to the absence of theoretical analysis, the lack of solutions to overcome the problem, the experiments on simplistic tasks and contexts, and issues on paper presentation. During the discussion period, the authors provided a new set of experiments on the Allen-Cahn equation, clarified the focus of their work, and changed the paper's structure and presentation. The reviewers acknowledged the improvements made in the revised version. After the discussion period, there was on accept, one leaning accept and two leaning reject recommendations.


The AE carefully reads the submission and discussions. He considers that the claims made in the submission are overall supported by evidence. The experiments are composed of a fair spectrum of problems: although some of them are simple, the added experiments on the Allen-Cahn equation strengthen the empirical validation. The submission addresses issues in the training dynamics of PINNs, which is of great interest for the community, and this work can certainly inspire follow-up works on theoretical aspects or more complex tasks.
The AE thus recommends paper acceptance under the request of a minor change about the validity domain of submission's claims. This could be included in section 4, and echoes the remarks made by reviewer KDWh and jz2A, e.g. to discuss if there are any classes of ODEs/PDEs where the assumptions made in the submission do not hold, making the training dynamics behave differently as it is expected in this work.

**Audience:**

The submission addresses issues in the training dynamics of PINNs, which is of great interest for the community, and this work can certainly inspire follow-up works on theoretical aspects or more complex tasks.

**Claims And Evidence:**

The claims made in the submission are overall supported by evidence. The experiments are composed of a fair spectrum of problems: although some of them are simple, the added experiments on the Allen-Cahn equation strengthen the empirical validation.

---

> ### Author Response · Authors · 2023-01-23
> **Camera-Ready Version Submitted**
>
> We have included the requested changes in the discussion section and uploaded the camera-ready version.
> Code to our experiments can now also be found on GitHub.
>
> We want to thank again the reviewers for their constructive advices and express our deepest gratitude to the AE for his thorough consideration and positive decision on our manuscript!